# A Qualitative Approach to Explore Perceptions, Opinions and Beliefs of Communities who Experienced Health Disparities towards Chronic Health Conditions

**DOI:** 10.3390/ijerph20085572

**Published:** 2023-04-19

**Authors:** Jacob C. Matos-Castro, Axel Ramos-Lucca, Ashley A. Rosa-Jiménez, Alessandra M. Beauchamp-Lebrón, Jorge L. Motta-Pagán, Luisa M. Morales-Torres, Eida Castro-Figueroa, Fernando J. Rosario-Maldonado, David A. Vélez-Maldonado, Dorimar Rodríguez-Torruella, Gloria Asencio-Toro, Melissa Marzán-Rodríguez, Julio Jiménez-Chávez

**Affiliations:** 1Psychology Program, Ponce Health Sciences University, Ponce, PR 00716, USA; 2Public Health Program, Ponce Health Sciences University, Ponce, PR 00716, USA; 3Psychology Program, Pontificia Universidad Católica de Puerto Rico, Ponce, PR 00717, USA

**Keywords:** chronic health conditions, community-based participatory research, focus group discussion, community needs

## Abstract

The prevalence of chronic medical conditions is associated with biological, behavioral, and social factors. In Puerto Rico (PR), events such as budget cuts to essential services in recent years have contributed to deepening health disparities. This study aimed to explore community perceptions, opinions, and beliefs about chronic health conditions in the southern region of Puerto Rico. Framed by a Community-Based Participatory Research (CBPR) approach, this qualitative study developed eight focus groups (n = 59) with adults (age of 21 or older) from southern Puerto Rico, in person and remotely, during 2020 and 2021. Eight open-ended questions were used for discussions, which were recorded, transcribed, and analyzed via computer analysis. Content analysis revealed four main dimensions: knowledge, vulnerabilities, barriers, and identified resources. Relevant topics included: concerns about mental health—depression, anxiety, substance use, and suicide; individual vulnerabilities—risk behaviors, and unhealthy habits; economic factors—health access and commercialization of health. Resource identification was also explored, and participants discussed the importance of alliances between public and private sectors. These topics were addressed across all focus groups, with various recommendations. The results highlight the importance of prioritizing identified community needs, evaluating available resources, and promoting tailored-made interventions to reduce risk factors for chronic health conditions.

## 1. Introduction

Chronic conditions are defined as long-lasting physical or mental diseases that either require prolonged medical treatment or conditions that limit activities of daily life or both [1,2]. Chronic medical conditions are a public health problem, considering that half of American adults have at least one chronic health condition and 27.2% suffer from multiple chronic conditions [3]. These conditions are the leading cause of death and disability, accounting for approximately USD 3.8 trillion in US healthcare costs annually [2,4].

In Puerto Rico (PR), for example, chronic physical and psychological conditions are highly prevalent [5]. The Department of Health of Puerto Rico reported diabetes, heart diseases, stroke, asthma, arthritis, and depression as the most prevalent chronic conditions on the island, the latter increasing as individuals age [6]. Puerto Rico’s vulnerabilities regarding chronic health conditions are linked to important health disparities that feed on governmental budget cuts in education, the healthcare system, and other Social Determinants of Health (SDH), such as over 40% of the population living under the poverty line [7].

## 2. Social Determinants of Health, Risk Factors, and Chronic Conditions

SDH are the parameters in which people develop their lives and age and the wider systems that shape daily life conditions [8]. These factors, while nonmedical, impact health beyond the effects of healthcare or lifestyle choices and account for 33–55% of health outcomes. In addition, SDH influence health inequities within and between countries. For Hispanics, the most prominent SDH affecting health outcomes are physical and social factors such as neighborhoods, housing, transportation, and employment conditions [9]. Health disparities in Puerto Rico are rooted in many factors related to the island’s history and culture, as well as its political status and economy [7,10]. The ongoing SDH, along with recent natural disasters and the current pandemic, exacerbate the effects of already existing health disparities [7,11,12] and worsen chronic health conditions. Whereas SDH shape health conditions and outcomes, risk factors contribute to SDH. Accordingly, risk factors and SDH occupy a bidirectional relationship, whereby one influences and is acted upon by the other.

Risk factors are elements that precede and are correlated with an increased likelihood of negative results at the biological, psychological, family, or cultural level [13]. Several factors influence the onset and progression of chronic diseases, such as genetic predisposition, SDH, prevention mechanisms, and risk behaviors. Physical inactivity, unhealthy diets, tobacco use, drug abuse, unprotected sexual practices, and harmful alcohol use are all significant health risk behaviors [14,15]. Furthermore, studies have revealed direct relationships between risk factors and certain chronic conditions. For example, one of the most researched relationships in epidemiology, including the absence of physical activity, is the relationship between cardiovascular diseases and diet [16]. On the other hand, diagnoses such as Alzheimer’s disease, which is linked to genetic predisposition by 70%, are more likely to be increased by other conditions such as obesity and hypertension [17].

Additionally, social factors contribute to the determinants of chronic medical conditions and have a direct link to risk behaviors. For example, smoking and secondhand smoke exposure contribute to the development of at least 21 chronic diseases, including at least 12 types of cancer, six types of cardiovascular diseases, diabetes, chronic obstructive pulmonary disease and pneumonia, and influenza [18]. Most recently, the Puerto Rican Department of Health developed a plan of action for chronic diseases from 2014 to 2020, in which they identified various risk factors and disparities within the population. The prevalence identified for each risk factor included the following: poor physical activity (66.2%), poor eating habits and overweight (39.8%), obesity (26.3%), high cholesterol levels (38.2%), hypertension (36.8%), and tobacco consumption (14.8%), which was consistent with those established by the World Health Organization (2013) [15].

An awareness of trends in SDH, as well as their multiple risk factors and impact on chronic health conditions, may allow health practitioners to build treatment regimens that evaluate patients with chronic diseases and identify where treatments could be primarily aimed [19]. The inclusion of an SDH view is needed for a better comprehension of how to provide better services to individuals with chronic conditions. Research suggests that cultural practices that are open to change and value quality improvement are necessary for implementing the tools and care methods, such as those described by the Chronic Care Model to address health risk behaviors [20]. Community engagement through focus groups that evaluate community health needs is a useful approach to better understand the unique needs of the community and is especially beneficial for the Puerto Rican population.

## 3. Community Engagement

Community-Based Participatory Research (CBPR) is an effective model to address community health needs and disparities at the community level [21]. The benefits associated with community engagement depend on trust and commitment between researchers and community members to share and interact as collaborators; moreover, they have implications in addressing difficult public health problems, such as chronic health conditions [22]. One way to engage community members is through focus groups, which allow community members to voice their opinion, beliefs, and concerns regarding their lived experiences related to chronic conditions. As a tool of community engagement, focus groups tend to be the preferred approach to evoke valuable information regarding sensitive topics when compared to individual interviews. Focus groups are also inclined to generate more themes about health problems that may not occur during individual interviews [23].

Community engagement approaches increase research relevance and efficiency and benefits the community, the investigators, and studies [24]. Plus, engagement builds trust among the community and improves local health services [25]. This research model has been demonstrated to be the superior approach in addressing health issues when compared to standard programs [26]. This is especially true when referring to mental health wellness among depressed individuals that are both older and economically disadvantaged. Outcomes regarding physical health and concerns of the cost of mental health services also improve with community engagement approaches. Preliminary interventions and studies have shown a significant improvement in mental health-related quality of life in low-income participants [26,27], suggesting that community engagement is a relevant and appropriate practice in impoverished communities. This study is an effort by the Ponce Health Sciences University (PHSU)–Research Center of Minority Institutions’ (RCMI) Community Engagement Core (CEC) which aimed to reduce health disparities associated to chronic medical conditions among the disadvantaged population in southern Puerto Rico. This paper reports one approach that the team is using to integrate community engagement in an attempt to identify perceptions, opinions, and beliefs in regard to chronic health conditions among low-income communities in PR.

## 4. Method

### 4.1. Study Design and Theorical Approach

We used an exploratory qualitative design using the Health Belief Model. Focus groups are an effective strategy for gathering sensitive and health information [23], and online focus groups increase participant diversity in terms of geographical location and other socio-demographic indicators [28,29]. Participants were residents of the municipalities of Ponce, Juana Díaz, Villalba, Santa Isabel, and Peñuelas. To be able to participate in the focus groups, participants had to be 21 years or older and able to provide informed consent. Those who had cognitive or physical impairments that would limit their understanding of the study, and participants who had been impacted previously by our CEC, would not be eligible to participate in the study. All participants provided informed consent, with all forms approved by the Institutional Review Board (IRB, protocol number: 1904011672R002) of the PHSU.

The CEC has established a team composed of community members and academic researchers. Community members play an active and participatory role through two groups: The Community Trained Workforce (CTW) and the Community Scientific Advisory Committee (CSAC). A series of meetings were held with both groups and the researchers; the CSAC participated in the development of the focus groups’ question guide to ensure the comprehensibility, sensitivity, and cultural relevance of the questions. The CSAC also provided feedback on the interpretation of the findings. The CTW collaborated on the design, the implementation of the recruitment strategy, and collaborated in the pilot test for the question guide with members of the community who met the characteristics of the target population.

### 4.2. Participants and Recruitment

A total of eight focus groups (n = 59) were composed of adults (21 years of age and older) from the southern region of Puerto Rico in two modalities, i.e., in person and remotely, during the 2020 and 2021 calendar years. Study participants were recruited through active collaboration with CTW members, local community leaders of the target towns (using word-of-mouth), social networks, and flyers focused on low-income communities of areas identified under the poverty level [30].

### 4.3. Data Collection

The focus groups began by in-person modality in November 2019; later, remote Zoom groups were included. The data collection process finished in June 2020. After signing the informed consent, participants completed a sociodemographic record sheet that included sex, age, monthly family income, their highest level of education, city of residence, and marital status. The focus groups were conducted by a facilitator (member of the research team with experience conducting focus groups) and two co-facilitators (trained research assistants) using the question guide. All sessions were held in community centers, private rooms of municipal facilities, and PSHU. The sessions were conducted in Spanish and each lasted 90 min. Questions that guided the groups’ discussion included the following:What comes to mind when you hear the phrase “chronic diseases?”What can you say about the chronic diseases present in members of your community or relatives?What do you think causes chronic diseases?

## 5. Qualitative Analysis

All focus group sessions were audio recorded using audio recorders and the Zoom platform. All of the audios were transcribed [31] and analyzed using the qualitative data analysis software, Atlas.ti version 8. The team conducted a content analysis using two theorical approaches, namely, Grounded Theory [32] and the Health Belief Model [33]. The content analysis supports the systematic interpretation of data to identify patterns, themes, and meanings [34]. The principles of Grounded Theory were used to openly identify and classify emergent topics from the predeterminant domains and proper topics from the Health Belief Model. To classify and organize the collected data, a thematic framework was developed according to the themes, concepts, and categories of the text [35]. Using this combination of theory and model, we aimed to conduct a deductive analysis based upon the different components found in the Health Belief Model, as well as any other sub-themes identified via coding.

A group of four analysts was convened to analyze the transcription and begin the categorization process. This process was divided in two rounds: The first was completed individually by research team members (A.R.-L., J.C.M.-C.), who then classified the findings into broad categories and sub-categories. All transcripts were then re-reviewed using the codebook. To ensure analysis reliability, two research team members (J.C.J.-C., M.M.-R.) coded the material separately a second time using the same process. Disagreements were discussed and resolved through an expanded codebook definition. Finally, the CTW and CSAC participated in the process of interpreting the obtained results to ensure an adequate and contextualized understanding of the phenomenon studied through the community’s unique vision.

## 6. Results

Most of the participants were female (86%), between the ages of 35 and 64 years old (22%), and mean age was 41 years old. Further, most participants were married/partnered (66%), had an Associate’s degree (22%), and reported a family monthly income of USD 1501.00 or more (37%). The objectives were addressed using eight open-ended questions. Table 1 describes the sociodemographic characteristics of the participants from all focus groups.

Content analysis revealed four main themes: knowledge about chronic diseases, vulnerabilities of communities for developing chronic illnesses, barriers to reduce chronic diseases, and identifying resources to address chronic medical diseases. Among these, sub-themes regarding mental health, individual vulnerabilities about risky behavior, health access, and resources found in the community were the most prominent. Table 2 shows all themes discussed in all groups and provides information regarding the themes identified in the study by the number of quotations associated with the theme.

### 6.1. Theme 1. Knowledge about Chronic Diseases

This theme addresses the participants’ knowledge of chronic conditions, including how these conditions impact their communities and/or relatives, as well as the wide and specific concerns these diseases may bring to people, the importance people ascribe to mental health, and the wide and specific concerns about this topic. Participants from nearly all focus groups defined chronic conditions as long-lasting diseases that cannot be cured, but that, through following their treatments, they could control them and prevent them from becoming worse. In many instances, participants also mentioned different kinds of illnesses their community was concerned about and used them as support for their definition of chronic illness. Several participants mentioned mental health conditions as their definition of chronic diseases. Participants also mentioned medical conditions of concern as part of the discussion. Other important themes for the groups included access to information, prevention strategies, and worries (see Table 2). Some participants showed concern in regard to being unable to obtain sufficient health information, as they believe there is a lack of interest in the community to learn about these diseases. One of the most relevant themes related to concerns about mental health, specifically depression, anxiety, substance use, and suicide. One of the participants stated:

“These illnesses have no cure, but with treatment we can continue living with the condition day by day.”

Another participant shared:

“As rheumatoid arthritis, you can get treatment to improve the quality of life, but these are illnesses that cannot be cured. You get treatment that can help you move and go on with daily life.”

Other participants mentioned mental health to support their definition of chronic illness:

“In my opinion, mental health conditions cannot be cured, you can improve with medication, but the condition is still there.”

One of the participants indicated:

“What kills you the most is the depression that leads to death, you go around it in your mind over and over and it is what leads you to destruction, it destroys you as a human being, as a person and you even make mistakes that many times you regret.”

Another participant from a different group expressed:

“We do not know the diagnoses, but at the moment with the earthquakes and stuff like that, everyone gets outside and one has the opportunity to talk with others. I’ve seen some signs or indicators that there may be depression or anxiety in old people and not because of the natural situations that we are going through, but due to the loneliness and illnesses they have and that no one is looking out for them, or the difficulty they have to be treated, or the fear they have of dying from these diseases, etc.”

In regard to substance use, a participant shared:

“At least in my community, there are many bars, which makes the social sphere difficult, socially and psychologically. Because many of these people are lonely, that’s where alcoholism or substance use comes from. These social factors are what hinders prevention.”

Finally, regarding suicide, a participant mentioned this public health issue as a community concern:

“Suicide rates are high. All of a sudden you learn that someone committed suicide and you ask yourself why.”

### 6.2. Theme 2: Vulnerabilities of Communities for Developing Chronic Illnesses

This theme addresses the participants’ perceptions towards possible causes of chronic diseases. These causes may be related to social, genetic, behavioral, access, habits, or any other perceived factors of chronic diseases. All groups discussed their concerns regarding individual, familial, and community vulnerabilities for developing chronic illnesses. Some participants discussed concerns regarding difficulties related to substance use, medication misuse, improving eating habits, sensitive health conversations (such as sexually transmitted diseases), and environmental vulnerabilities.

One participant stated:

“Chronic conditions are caused by the abuse of alcohol and drugs. Also, what we eat. We don’t eat the way we should eat. For example, people with diabetes cannot eat some things and at times they make exceptions and at the end we fall into a pattern of behaviors that we do what we want without thinking of the consequences that will come after. In the case of HIV, most of it is due to the abuse of substances and also due to having sex without knowing the HIV status of the other person, etc.”

Another participant shared:

“Many also misuse medication. I can say: “I got a headache,” [and] someone comes and tells me they have a good pill for that and then I take it, but that pill was not meant for my use. It was not prescribed to me.”

In another group, a participant commented:

“The perception about alcohol in my city and community is that we are a place for alcohol consumption. Culturally, we are known for our collective drinking behavior. I think that is very significant among the teenagers. There are too many adolescents with beer cans in their hands and from what I can see, parents are open to the idea, as the mentality is that before their children get alcohol from someone else, they will give it to them. They see it as a means for trust between parents and children.”

### 6.3. Theme 3: Barriers to Reducing Chronic Diseases

This theme addresses the participants’ perceptions about receiving information on chronic diseases and community prevention, as well as barriers to acquire this information and the possible factors of perceived difficulty in accessing this information on a personal or community level. Most groups discussed individual factors, socio-structural factors, and economic barriers to reducing chronic diseases. Some participants mentioned these themes when discussing economic factors affecting health access and commercialization of health:

“Depending on your health insurance (another participant: “The economic factor”), the HIV issue, for example. At least my doctor sent me to do the test in the last general check-up and I asked him why because I’m not sexually active, I am a widow. He told me that my health insurance requires that it’d be done even to elderly as it is preventive medicine. There are many things that we are not aware of, but the resources are available for certain people. Some health insurances cover, others don’t.”

Another participant shared:

“Some people lack the resources to obtain their medication. Not everyone can cover their medication and we know of people in our community that cannot get their meds, as they cannot cover them.”

A participant in another group said:

“I remember [provider name] that was located on Villa Street, they would screen you for cancer annually if you were a woman. They would monitor you. But now you need to make a request to the doctor for a cancer screening. In my family, we have genetic heritage and sometimes I have vaginal pain or whatever and you say that to the doctor and they start saying: “No. You don’t need any screening because you have one from 2016 and you were good. But what about 2017? I do not know. I don’t know if I have cancer right now.”

### 6.4. Theme 4: Identified Resources to Address Chronic Medical Conditions

This theme addresses resources related to chronic disease prevention that people identify based on their experiences or knowledge, along with their perceptions of what is useful or not for prevention purposes. This theme also comprises descriptions of how participants find these resources to help prevention. Moreover, this theme includes participants’ opinions on what they consider important about chronic illnesses and their preventions, as well as the benefits, including perceptions about activities aimed to prevent chronic diseases and whether they consider these beneficial or not. Some participants discussed individual and community resources to address chronic medical conditions and stated the importance of alliances between public and private sectors:

“I think that the universities should make an alliance with the government and bring those health professionals that they are preparing and develop clinics in different towns, as it would be of help. For example, there are municipalities that spend a million dollars on a health bus and it has everything, but what good is it if it’s always parked and not taken around the neighborhoods or to the people in need?”

Another person commented:

“One of the resources I know is the group of nurses that come to do their practicum with us in the community. They also give us health information.”

In another group, a participant shared:

“In my experience with coordinating health and education fairs in my community, we invite different agencies from Departamento de Salud, WIC to different health insurances and universities. We also invite organizations of different patients and we offer all kinds of health services. But I still think we should develop a way to provide a more consistent and personalized experience for the participants, to provide a referral and to provide follow ups.”

All four themes were addressed in every focus group (n = 8) along with a variety of recommendations to improve chronic health conditions in the community:

“It is not the same to impact a population of the elderly, adolescents, or children. You cannot impact them with the same brochure. You have to know what their interests. Teenagers right now, they like ‘reggaeton’ (latin hip-hop), they like to hang out, to have fun. Maybe getting people relate to that. You sang it, you learned it. What you read and study, you forget. You have to try to find what people like.”

Another participant recommended,

“We, residents of public housing, are required to do community work. I think it should also be implemented that we as residents of public housing have to train ourselves about different diseases in order to help. Thus, making a positive collaboration to society because the reality is that we are not doing it. Many times, we are not interested. Other times we are interested, but we do not have an extreme need until we reach the point that it is affecting us personally, and then we want to know, when we can no longer do anything, because well, you already have the problem at hand. I understand that they (agencies) should facilitate it financially because many times those classes are too expensive and other times you know that this artifact (cardiac defibrillator) is there, but you don’t know how to use it.”

## 7. Discussion

Focus group discussions promote a conversation about *chronic health conditions* such as cancer, heart diseases, and diabetes, accompanied by themes regarding mental health issues [23]. The most mentioned topics, *barriers* and *vulnerabilities*, reflect the multiple challenges that participants faced in managing their chronic diseases in the context of multiple public health emergencies (hurricanes, earthquakes, the ongoing pandemic) and socio-medical structural crisis in Puerto Rico. Supported by the results of this study, socio-structural factors are the main challenge for promoting the better management of chronic diseases in Puerto Rico. Even when *individual* (lifestyles factors) and *interpersonal topics* (family challenges) were identified during the analysis process, the most highlighted challenges mentioned by participants were those related to the difficulty of navigating the *healthcare system* and the *economic* struggles to overcome certain *barriers* (transportation, income, access to medication, healthy food). These results highlight the critical role of the SDH framework as part of the Puerto Rico healthcare system.

The results suggest the importance of prioritizing identified community needs, evaluating their available resources, and promoting tailored-made intervention models to reduce risk factors associated to chronic health conditions. Interventions to improve health outcomes must consider the reality of communities related to SDH, such as transportation, navigating a fragmented health system, and the limitations of access to mental health services. Our experience reflects that the use of the approaches of community engagement, fostering the active participation of community members in different phases of the needs assessment process (for example, development of guiding questions, recruitment of participants, interpretation of outcomes), is a helpful strategy to gain a better understanding of SDH-related factors that act as barriers that perpetuate poor community health outcomes.

Finally, the results of this analysis have been used to inform: (1) the different educational materials to address some areas identified as having a “lack of information of chronic disease awareness and management”; (2) quantitative survey using the relevant topics of these results; (3) the semi-structured guide for the next qualitative round of focus groups. This study will be succeeded with a quantitative component to provide an informed and community-based intervention focus on SDH and chronic health conditions in Puerto Rico. In addition, this manuscript is a vehicle that allows the dissemination of the voices of the community that know their needs and that, in many scenarios, has shown its willingness to be an agent of change to improve its social realities.

### 7.1. Future Research Strategies

As part of the learning lessons for the research team, the pandemic was an opportunity to increase the reach of participants, including people with mobility limitations and those who were bedridden. Although participants were unable to attend in-person focus group sessions, this opportunity provided the space integration of key populations to deeply explore their experiences of health disparities. Finally, the team continues to support the use of the qualitative method as an outstanding tool to continue strengthening health disparity research.

### 7.2. Limitations

In the context of the COVID-19 pandemic, due to the restrictive measures imposed, authorization from the IRB was requested to carry out the groups remotely via the Zoom platform. Though it was not planned, this challenge did not represent a methodological issue for analysis purposes, which is a strength identified throughout the study and an opportunity for future ones.

When conducting focus groups during the beginning of the COVID-19 pandemic, it was a challenge to assure that, under a community engagement approach, the research team would be able to guarantee access to participate in these groups. Through the team’s community infrastructure, supportive community leaders helped find potential participants in the targeted municipalities and invited them to engage in a virtual modality to complete the study.

## 8. Conclusions

This study explored current community perceptions and knowledge regarding chronic health conditions in Puerto Rico. Results show that many participants identified current barriers for the prevention, management, and education of these topics. This shows the need to create community oriented interventions that answer the needs of the population. This study has shown that, currently, there are various health topics in need of not only further research studies, but also effective interventions to increase overall health and reduce risk factors. A multidisciplinary approach should be considered, while also making sure that the communities and their respective community leaders are involved in the creation of meaningful partnerships which, in turn, can create better opportunities for bettering health outcomes.

## Figures and Tables

**Table 1 ijerph-20-05572-t001:** Focus groups sociodemographic characteristics.

Sociodemographic	n	%
Age (years)	59	
21–24	3	5
25–34	12	20
35–44	13	22
45–54	7	12
55–64	21	36
65+	3	5
Sex	59	
Female	51	86
Male	8	14
Highest education level	58	
Less than high school	6	10
High school	14	24
Technical degree	7	12
Associate’s degree	13	22
Bachelor’s degree	8	14
Master’s degree	7	12
Doctor’s degree	3	5
Marital status	58	
Single	13	22
Married/Partnered	39	67
Divorced/Separated	5	9
Widowed	1	2
Family monthly income	57	
Less than USD 500.00	11	19
USD 501.00–1000.00	15	26
USD 1001.00–1500.00	10	18
USD 1501.00 or more	21	37

**Table 2 ijerph-20-05572-t002:** Content analysis of the focus groups.

Category	n Number of How Many Focus Groups Mentioned the Category	Number of Quotations Associated to Each Theme
I. Knowledge		-
I.a. Definition	7	29
I.a.a. Types of chronic illnesses	-	-
a.1 Mental Health	8	68
a.2 Cardiovascular health	8	28
a.3 Neurological	5	15
a.4 Metabolic disorders	8	36
a.5 Rheumatologic	3	7
a.6 Infectious	8	16
a.7 Cancer	8	40
a.8 Psychoneuro-inmunology	6	16
a.9 Respiratory	5	13
a.10 Other	8	43
I.b. Access to information	6	54
b.1 Benefits	8	26
b.2 Barriers	8	39
I.c. Prevention strategies	8	28
c.1 Primary	7	24
c.2 Secondary	7	20
c.3 Tertiary	2	2
I.d. Worries	1	3
d.1 Chronic illness in children/family	5	15
d.2 Sexually Transmitted Diseases	1	1
d.3 Access to professional help	7	15
d.4 Economic barriers	3	8
II. Vulnerabilities	-	-
II.a. Individual	8	71
II.b. Family	8	45
II.c Community	8	50
III. Barriers	1	1
III.a Individual	8	61
III.b Socio-structural factors	7	30
b.1 Economical	8	63
b.2 Political	3	4
b.3 Access to technology and communications	3	10
b.4 Service access	3	8
IV. Identified resources	1	-
IV.a Support	2	2
a.1 Individual	6	12
a.2 Family	3	9
a.3 Community	8	57
IV.b Recommendations	8	79

Note: A total of eight focus groups were conducted with community members (n = 59).

## Data Availability

Data available on request due to restrictions e.g., privacy, or ethical. The data presented in this study are available on request from the corresponding author. The data are not publicly available due to private information that could identify participants, therefore we must verify that any data shared does not have any quotes that could identify participants.

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
