# Peer review of "A Qualitative Approach to Explore Perceptions, Opinions and Beliefs of Communities who Experienced Health Disparities towards Chronic Health Conditions"

_ijerph, 2023, doi:10.3390/ijerph20085572_

Round 1

Reviewer 1 Report

I enjoyed reading the manuscript.  I only have a few suggestions:

1. Regarding Table 2, it would be helpful to explain what the N is (the number of groups that mentioned the topic?). Also, in the second page of the table 2 there seems to be some spaces created by numbers that moved down. Please make sure the numbers in each line are correct.

2. In the result section make sure that all the theme's tittles are written in Italics.

3. In the discussion you mention the importance of "promoting tailored-made intervention models to reduce risk factors associated to chronic health conditions."  I would like to see a little more elaboration about how this could be done, since it has important implications for public health practice.

Author Response

  1. Table 2 has been changed to include the recommended title
  2. Theme Titles have been italized
  3. A new paragraph has been added to elaborate on the section mentioned

Reviewer 2 Report

This is a review of the manuscript entitled, "a qualitative approach to explore perceptions, opinions, and beliefs of communities who experienced health disparities towards chronic health conditions" [ijerph-2219471].  It seeks to explore themes raised by community members regarding their knowledge and lived experiences with chronic illnesses within specific municipalities on the island of Puerto Rico.  For major themes are identified in this qualitative analysis of focus group data conducted during the COVID-19 pandemic. The following comments are in an effort to help improve the manuscript in its current form.

  Page 2, A good deal of time is spent talking about social determinants of health and their relationship to risk factors.  However,  it is less clear how this discussion informs the research question. Please provide more clarification for how the specific research methodology, combined with the SDH and risk factor conversation help the authors answer the research question(s).

page 2, line 85-94. Please clarify the rationale for the current study, in other words, what are the knowledge gaps or unknowns that prompted this line of inquiry? The statement, "Awareness of trends in SDH...may allow health practitioners to build treatment regimens..." is a good start, but why are focus groups necessary to fill knowledge gaps. Please make more of a case for why focus groups vs. quantitative methods or other epidemiologic methods are called for to get at the research question.

page 6, line 207. "...but can improve their daily lives..." did the participants begin to talk about chronic disease related growth? If so, this would be worthy to explore in more detail.

page 8, lines 304-307. Sentence starting, "In addition..." is confusing. Not sure the point the authors are trying to make

Discussion

pg 9, line 356. It may help the manuscript if the authors could tie the results back to a model of care or demonstrate how this research filled an important gap in knowledge or will propel future research to intervene on health disparities identified by participants.

It is clear that COVID-19 and the resulting pandemic presented challenges for conducting research. In the Future Research Strategies, please devote more time to how the current research will inform future studies the team plans to conduct. I would suggest placing the COVID-19 challenges into a separate section labeled limitations.

  Overall, the reader is left with the question, how do the findings lead to more effective interventions or tailored care as was raised in the introduction to the manuscript. There seems to be a disconnect between the research methodology, results discovered and translation into tailored interventions that try to decrease risk factors in the community.

  The authors should be commended for sharing the themes that were discovered during the focus groups, it is always powerful to hear participants voices come through a scientific manuscript.   It is clear they wanted to understand the needs of the community and it will be exciting to see how this translates into  tailored medical interventions.

Author Response

Responses to Comments:

  1. A new sentence has been added in lines 89-90
  2. The use of focus groups was considered due to our desire for Community engagement; we intended to work closely with community partners to obtain input from community leaders and members who know the needs of their communities. The use of focus groups is discussed in the 3rd page, lines 102-108 “One way to engage community members is through focus groups, which allow for community members to voice their opinions, beliefs, and concerns regarding their lived experiences related to chronic conditions. As a tool of community engagement, focus groups tend to be the preferred approach to evoke valuable information regarding sensitive topics when compared to individual interviews. Focus groups are also inclined to generate more themes about health problems that may not occur during individual interviews”
  3. Participants, in this case, spoke of how, following treatment plans, they can control chronic health diseases and potentially prevent worse outcomes; the line has been changed to reflect what was said.
  4. Sentence has been changed
  5. A sentence has been added to answer the recommendation
  6. Limitation section has been added
  7. We have included a new paragraph to better tie in the idea of interventions to improve health outcomes through the use of active community participation

Reviewer 3 Report

Dear authors,

Thank you for submitting this paper. Overall, I find the study well designed and the paper presents a concise and well written study. I have some suggestions for revision.

In the method section, there is some statements that would better fit in a limitations- och methodological considerations section within the discussion. The method section should only report on how you did- not motivate these choices (such as in page 3 line 129-130)- move this to the discussion section.

I also suggest you to use the headings study setting, study sample and participant recruitment or similar to structure the method section and make it easier for the reader to see who and how the study participants were recruited and included. Also clearly state the inclusions and exclusion criteria used to identify study participators.

As far as I understand, you have used some kind of mix of thematic analysis, content analysis and Grounded Theory in your study? I find it very strange to do that way, since all of these are specific, rigor but very different qualitative methods. I suggest that you focus on one method and follow that in accordance with the specific method you have used, and refer to that method throughout the whole study.

The results are a bit depending on what method you use. To me, this is some kind of mix between quantification of qualitative data, as suggested for example by Sandelowsky, when you specify the number of quotation and so on. Otherwise, it seems more of a manifest content analysis (such as presented in the Graneheim & Lundman content analayis methodology for example).

As suggested above, I also suggest that you include a section of methodological considerations and limitation in the discussion section.

Author Response

  1. Methodology section was reviewed, and remote groups as a limitation were removed and added to the limitations section
  2. Inclusion and exclusion criteria have been expanded to be included in the methodology section, and suggested Titles were implemented in the manuscript
  3. Our study has an exploratory quantitative design that uses The Health Belief Model and Grounded Theory to identify patterns, themes, and analysis. Using the principles of Grounded Theory, we were able to establish topics and themes that can be found in the Health Belief Model. The first line has been changed to reflect this,

Round 2

Reviewer 3 Report

Dear authors, 

Thank you for the revised version of this manuscript. I still do not really understand the combination of qualitatiev methods used. I guess you have used an deductive analysis methodology, but this is very uncelar. I leave for the editor to decide on how to preceed. Otherwise, I find all my suggestions well met. 

Author Response

We thank you very much for your recommendations and review

We have added sentences to lines 183-186 to clarify the methodology

Otherwise we will take your advice and leave it to the editing department. Thank you very much!